# Bioinformatics: From NGS Data to Biological Complexity in Variant Detection and Oncological Clinical Practice

**DOI:** 10.3390/biomedicines10092074

**Published:** 2022-08-24

**Authors:** Serena Dotolo, Riziero Esposito Abate, Cristin Roma, Davide Guido, Alessia Preziosi, Beatrice Tropea, Fernando Palluzzi, Luciano Giacò, Nicola Normanno

**Affiliations:** 1Cell Biology and Biotherapy Unit, Istituto Nazionale Tumori—IRCCS—Fondazione G. Pascale, 80131 Naples, Italy; 2Bioinformatics Research Core Facility, Gemelli Science and Technology Park (GSTeP), Fondazione Policlinico Universitario Agostino Gemelli IRCCS, Largo A. Gemelli, 8, 00168 Rome, Italy

**Keywords:** NGS, variant calling, cancer, biological complexity, ML/AI algorithms, network analysis, homologous recombination deficiency, targeted therapy, big data

## Abstract

The use of next-generation sequencing (NGS) techniques for variant detection has become increasingly important in clinical research and in clinical practice in oncology. Many cancer patients are currently being treated in clinical practice or in clinical trials with drugs directed against specific genomic alterations. In this scenario, the development of reliable and reproducible bioinformatics tools is essential to derive information on the molecular characteristics of each patient’s tumor from the NGS data. The development of bioinformatics pipelines based on the use of machine learning and statistical methods is even more relevant for the determination of complex biomarkers. In this review, we describe some important technologies, computational algorithms and models that can be applied to NGS data from Whole Genome to Targeted Sequencing, to address the problem of finding complex cancer-associated biomarkers. In addition, we explore the future perspectives and challenges faced by bioinformatics for precision medicine both at a molecular and clinical level, with a focus on an emerging complex biomarker such as homologous recombination deficiency (HRD).

## 1. Introduction

Genomic profiling has assumed an increasing role in the clinical management of cancer patients, thanks to the approval of numerous drugs showing demonstrated activity in patients with specific genomic alterations [1,2,3,4]. The need to identify an increasing number of complex genomic biomarkers led to the introduction of next-generation sequencing (NGS) technologies in clinical practice. The massive amount of data generated by NGS experiments required the development of algorithms based on computationally efficient statistical methods and artificial intelligence, to improve the processes of genomic variant detection, visualization, and interpretation in terms of pathogenicity [5,6,7,8]. The implementation of bioinformatics approaches has a clear intention to elaborate and process data in an efficient and fast way to avoid turnaround times and increase detection accuracy [9,10,11,12,13]. Although new computational solutions have gradually been proposed, showing incredibly high levels of accuracy, many issues in clinical practice remain unsolved. The most important of these challenges is represented by our ability to interpret the potential impact of genomic alterations on a patient’s health, and how this information can be used to tune personalized therapies [14,15]. In this review, we will describe some important technologies, computational algorithms and models that can be applied to NGS data ranging from Whole Genome to Targeted Sequencing to address the biological complexity of cancer. Finally, we will explore the future perspectives and challenges faced by bioinformatics for precision medicine both at a molecular and clinical level, with a special focus on Homologous Recombination Deficiency (HRD) as an emerging biomarker in clinical practice.

## 2. NGS Technologies and Bioinformatics Tools for DNA Sequencing Data

### NGS Approaches in Cancer Patients’ Management

Different sequencing approaches have been recently used to analyze the genomic landscape of tumors [16]. Traditionally, Sanger sequencing based on electrophoresis and involving the random incorporation of chain-terminated dideoxynucleotides by DNA polymerase during in vitro DNA replication requires time-consuming analyses on a single DNA fragment [17]. On the other hand, NGS can sequence millions of fragments simultaneously per run during automated cycles of synthesis, scanning and washing, playing a critical role in reading the mutational landscape of a patient in clinical routine. This process, applied to hundreds to thousands of loci per subject, generates an enormous amount of sequence data, with many possible applications in research and diagnostic settings including sequence variation detection (referred to as “variant calling”), epigenetic and transcriptional regulation, chromatin conformation, its 3D architecture, and how these phenomena influence each other [18,19]. This review will focus on variant calling and its preprocessing stages.

The primary output of NGS data analysis is the alignment of small DNA or RNA fragments (known as “reads”) using either a pre-assembled reference genome sequence (or “assembly”) or concatenating input reads using “de novo” strategies, in which no reference sequence is used. Diverse approaches are optimized for genomic analysis at different scales: from a few genes to the whole genome [20,21,22].

Whole Genome Sequencing (WGS) covers the whole genome, and it is used to investigate previously undescribed genomic alterations, requiring more time and higher costs [23,24]. Whole Exome Sequencing (WES) may cover protein-coding genes only, representing 3% of the whole genome, but with reduced costs and under the assumption that protein-associated alterations have often a deleterious impact on genome regulation [25,26,27,28]. Still, the complexity of data interpretation limits significantly the use of WES in clinical research and practice. Consequently, Targeted Sequencing (TS) was introduced for analyzing specific mutational hotspots, for a given genome [29,30,31]. This approach is used to detect disease-causing genomic alterations with described or suspected pathogenicity [32,33,34,35,36].

The typical NGS workflow is divided into several steps, including sample preprocessing, library preparation, sequencing and bioinformatics analysis. Each step plays a critical role and might hide sources of error that could propagate to the final output. Currently, variant detection and annotation are well-standardized procedures, performed through the following steps [37,38,39] (Figure 1).

A typical variant calling workflow involves the following data preprocessing steps: (i) quality filtering of the raw FASTQ files; (ii) read alignment through either reference-based or de novo alignment (BAM or CRAM files); (iii) duplicate reads removal from the alignment and mapping quality filtering; and (iv) local realignment and/or haplotype determination (phasing) [40,41,42,43].

Modern variant callers usually take BAM files from read aligners as input and perform only the last preprocessing step. As explained in Section 4, recent deep learning-based callers also include image-like alignment preprocessing (image pileup) and chromatin conformation analysis. Although the specific variant detection methods may vary for different algorithms, variant calling methods can be grouped on the basis of the input sample origin (germline or somatic), and by variant type: small nucleotide variants (SNVs); small insertions/deletions (indels); and structural variants, including copy number variants (CNVs) and large genomic rearrangements (such as insertions, deletions, and translocations) [44,45,46,47,48,49].

## 3. From NGS Data to Variant Discovery

### 3.1. Variant Types Relevant for Precision Oncology

The type of alterations that must be reported in clinical diagnostics and that might have a relevant clinical value include Single Nucleotide Variants (SNV), small insertion/deletion (InDel), Copy Number Variants (CNV) and genomic rearrangements that can lead to gene fusions. In addition, cancer-related complex biomarkers, such as Tumor Mutational Burden (TMB) and Microsatellite Instability (MSI) and HRD, are becoming frequently included in clinical reports for their clinical value.

### 3.2. Variant Discovery Workflow

The variant detection workflow is a sequence of steps, which include the sequencing quality control, the preparation of data (pre-processing) and the use of algorithms able to detect the genomic alterations. To date, the tools used for the variant discovery on tumor samples include three main steps: preprocessing, calling of variants, and annotation.

### 3.3. Sequencing Quality Control

A flexible, robust and most used tool in quality control is FastQC (https://www.bioinformatics.babraham.ac.uk/projects/fastqc/ accessed on 12 July 2022), developed at the Babraham Institute to examine sequencing quality, starting from fastq files. It works on all Operating Systems (OS) and can be used with both GUI interface and command line.

This option is commonly used by bioinformaticians, to add the quality control step in a custom pipeline. The latest versions of FastQC include Picard (https://github.com/broadinstitute/picard/ accessed on 12 July 2022), a tool developed by the Broad Institute to manage SAM, BAM, and VCF files, and to perform the quality control at different steps of the bioinformatics pipeline. A fast and simple tool to calculate the coverage starting from BAM files is Mosdepth [50]. It may calculate the coverage depth for both whole genome and exome sequencing data. It is also able to limit the analysis to a specific genomic region providing a bed file. This application could be useful also for targeted sequencing, especially for custom panels.

### 3.4. Pre Processing

The data preparation is described in the GATK Best Practices [51] and it is a compulsory step in order to provide the correct input to the variant detection algorithms. Many tools are used to provide the alignment, and to ensure the management of the duplicates and the recalibration phase. The final output is a BAM file ready to be analyzed for the variant’s identification. The preprocessing step has been consolidated over time thanks to The GATK Best Practices, developed by Broad Institute, and implemented in the GATK ToolKit [52]. This procedure allows us to produce the alignment files in the best possible way to investigate the presence of alterations in the sequenced genome.

### 3.5. Variant Calling

The variant calling process is the main step for DNA alteration discovery. It includes different algorithms able to find potentially pathogenic mutations across the human genome. The College of American Pathologists and the American Medical Informatics defined a list of 17 recommendations for clinical NGS bioinformatics pipelines [7]. These statements include, but are not limited to: (i) the involvement of medical personnel, (ii) stage design, (iii) version control, and (iv) reproducibility.

Quality control of each step of a bioinformatics pipeline is crucial to setting optimal parameters, achieving the best possible variant calling performance, and passing the validation step using a representative set of known variants across the samples. Special attention must be given to software versioning and data integrity, in order to track any changes/updates and prevent loss of information/data.

The validation procedure involves variants selection, sequencing quality control, algorithms, filtering and annotation. While the variants selection is a process carried out by scientists with different skills and training, the analysis performed in the sequencing data quality control are consolidated. Concerning the variant calling, there are as many tools as there are alterations to investigate. The main difference is between germline and somatic variants. Although the available algorithms are quite different, due to the intrinsic difference between germinal and somatic variants, the Input/Output file formats are the same: input BAM file(s) and output VCF file(s). The critical point in the variant calling is the filters applied to prevent false positive and false negative events. This depends also on the sequencing coverage/depth and the length of sequenced genomic DNA (e.g., panels vs. exome).

Although many SNV/indel detection algorithms have now reached high accuracy in several benchmarking tests [53], combining the results of multiple algorithms may increase sensitivity, thus reducing the rate of false negatives [54]. Different tools, such as BCFtools [55], enable the merging of multiple calls into a single VCF file. However, clinical practice generally requires clear and portable workflows, generating reproducible results. Furthermore, standard clinical procedures should encompass multiple variant types (SNVs, indels, and structural variants) and genomic alteration measures, including LOH and MSI, readily usable for targeted therapy. To this end, a huge effort has been provided by the nf-core group [56] through the Sarek pipeline [50] for germline and somatic variant detection. Although these tools often use different methods, they generally achieve comparable accuracy. The Sarek pipeline uses all the tools shown in Table 1, depending on the purpose (germline or somatic variant call) and the variant type (small variants or large genomic rearrangements).

### 3.6. Variant Discovery Pipelines: Tools and Algorithms

In this scenario, commercial and open-source pipelines are often integrated to provide efficient and customizable solutions. One of the players in the development of these tools, offering licensed software, is the Illumina^®^ company. Illumina^®^ adopted the GATK best practices as a consequence of a partnership with the Broad Institute. This produced a series of Illumina licensed and Broad open-source tools derived, available on the Broad Institute repository (https://broadinstitute.github.io/warp/ accessed on 12 July 2022). Both licensed and open-source solutions are released as “only for research” software and need to be validated by the institutions adopting them. The advantage of licensed software consists of its “ready-to-use” design. In fact, it does not need additional software and database installation procedures, thanks to container technology. On the other hand, licensed software offers less flexibility during the analysis flow than open-source tools. This limitation arises when, starting from raw data (usually, bcl or fastq files), they run almost uninterruptedly through preprocessing, quality control, variant calling, biomarker analysis (including MSI, TMB), and reporting. The consequence is that the whole analysis cannot be split for custom pipeline development and integration with other (possibly newer and more efficient) resources. In addition, combined output reports from licensed software often include only minimal SNV information, are uneasy to read for clinicians, and require further annotation and processing of the vcf files. Consequently, Illumina^®^ offers commercial solutions to produce clinical reports that make use of collaborations with other companies. Conversely, open-source solutions enable user control over each analysis step, allowing single analyses to run as independent modules. By definition, open-source tools need clinical validation to be adopted in diagnostics.

Recently, the nf-core community aimed at defining standard procedures to be included within bioinformatics analysis workflows. The goal of nf-core is to adopt the best practices in bioinformatics pipeline development, through an open-source and peer-reviewed community, in order to offer a reproducible, portable and robust solution in different fields of application. In this context, the Sarek pipeline [68] was developed for germline and somatic variants calling (Figure 2).

Sarek is designed to analyze data from WGS, WES, and TS. It allows us to perform the analysis starting from several intermediate steps, such as preprocessing or variant calling. In addition, the tools included in the pipeline workflow several variant calling types: germline, somatic, and tumor-only somatic. Consequently, researchers can break the analysis at any step, and, taking advantage of object-oriented programming, every result is an object that could be reused for separate analyses. Thanks to this philosophy, several analyses or quality controls are included and may be added in a continuous process of integration. As a result of this flexibility, the end user can either run the pipeline in a default mode, using a single command line or build a complex job, by using different tools as independent objects in several pipelines. Both nf-core Sarek and GATK are developed as object-oriented software, where the main difference is that the nf-core community allows to include other validated and peer-reviewed algorithms, with the advantage of complete reproducibility.

### 3.7. Annotation

Variant annotation is the process of assigning information to DNA variants and evaluating their possible pathogenicity. The annotation step is another crucial point and it is the bridge between machine and human-readable format. Furthermore, a punctual annotation of the variants is strongly linked to the query of updated databases. To this end, different open-source and licensed solutions offering reviewed and updated annotations, as VEP [69], VarSome [70], ClinVar [71], OncoKB [72], have been developed.

## 4. Machine Learning Applied to Next-Generation Sequencing and Variant Discovery

### 4.1. Sequencing Technology Issues and Machine Learning

The widespread of bio-medical machine learning (ML) methods followed the evolution of high throughput sequencing (such as NGS) technologies. This opened up a new class of diagnostic tools, drug discovery methods, novel patient stratification approaches, and personalized therapies. Besides the large amount of publicly accessible NGS data from consortia [73], the availability of low-cost sequencing platforms caused a worldwide growth of in-house data, and the consequent increasing demand for computationally-efficient, yet accurate and reusable, ML-based software. One of the toughest challenges tackled by ML in clinical genomics is to model what (and possibly how) genomic variants and their interactions influence cell development and fate, leading to cancer transformation [74]. Although classical inference methods can be highly flexible and interpretable in terms of causality, they are often constrained by linearity or model-based assumptions, providing aggregate (e.g., averaged) or partial descriptions of the underlying biological processes (e.g., neglecting nonlinear systems properties) due to a strong dependence on current knowledge or field-expert validation. Traditional methods, such as GATK [75], make extensive use of different statistical models and heuristics based on calling quality, allele frequency, and sequencing coverage, to estimate the likelihood of variation at each genomic position. However, this task is severely hampered by the presence of sequencing artifacts deriving, for instance, from polymerase chain reaction (PCR) errors, DNA synthesis dephasing and inefficiency, and low-complexity or repetitive genomic sequences, that are only partially manageable [76,77]. Moreover, sequencing data is inherently high-dimensional (i.e., the number of observed variants is way bigger than the number of sequenced genomes), the same phenotype can be caused by different combinations of variants (heterogeneity), and often only a few individuals carry the variation associated with the observed disease (sparseness). In addition, a phenotype rarely arises due to the presence of a single deleterious variant, but rather from (hierarchical) interactions among variants with marginal effects [78].

Collectively, these issues motivated the dissemination of ML algorithms, for several reasons. Firstly, their ability to learn patterns of interactions directly from data and, secondly, the capability to model complex hierarchical and nonlinear interactions without specific statistical modeling assumptions. The main costs for these advantages are the need for very large labeled training sets (i.e., supervised learning), with an obvious impact on the computational demand, and potential vulnerability to the training set compositional biases, often caused by incomplete or imbalanced knowledge of specific domains [62].

### 4.2. ML and Deep Neural Networks for Variant Discovery

Given their capability of modeling a very large number of features and parameters, deep neural networks have been extensively used for variant discovery. The main idea beneath convolutional neural networks (CNNs) is to convert pileups of aligned reads into patterns of an image, resulting in groups of interconnected variants that might have a pathogenic effect [79].

CNN-based algorithms include the general-purpose software DeepVariant [62], the specialized Clairvoyante [79], NeuSomatic [80], and DeepSV [81], designed for single-molecule technologies, somatic variants, and structural variants, respectively. In many cases, improvements in predictive accuracy have been achieved using ensemble methods, where the learning process takes advantage of different integrated models. For instance, CNNScoreVariants [82] exploits pre-trained models in GATK to discover SNV and indels from short-read sequencing data.

The Clair algorithm [79], the successor of Clairvoyante, uses CNNs in combination with recurrent neural networks (RNNs) and feedforward networks to refine germline SNV and indel discovery. While these methods generally outperform traditional inference methods for SNVs and indels, much less effort has been spent on the more complex structural variants (SVs). To this end, DeepSV [83] uses CNNs to find large (>50 bp) genomic rearrangements, including insertions, deletions, and inversions.

One main limitation of many deep learning algorithms resides in possible information biases within their training sets [84]. The goal of variant discovery is to find genomic loci that are causally associated with the disruption of one or more molecular functions and pathways. For coding DNA, pathogenic alterations are likely to alter the structure and function of the encoded protein and therefore are much easier to be associated with a diseased phenotype. Accordingly, most of the diagnostic procedures in clinical and cancer genomics are either based on panels of a limited set of exons or WES [85,86,87]. However, many of the deleterious traits of disease are caused by noncoding variants that are likely to be located at regulatory elements [87].

To cope with this possible bias, the DeepSEA [88] and Basset [89] algorithms use CNNs to predict the chromatin state and chromatin accessibility that may reveal the presence of regulatory elements. Both DeepSEA and Basset learn the regulatory sequence code from genomic sequence by training a deep CNN over large chromatin-profiling data from ENCODE and Roadmap Epigenomics consortia. These data include transcription factor binding, DNase I sensitivity and histone-mark profiles. Learning from data-driven features rather than annotations (e.g., exons) allows these algorithms to detect noncoding variants with a possible regulatory role. In addition, the deep neural network structure allows us to scale on sequence length, enabling the use of large contextual genomic regions and further improving noncoding variant function interpretation. An alternate application of these methods consists in validating and converting ML results in current knowledge, improving and speeding up current clinical trial protocols, providing tools for efficient (low attrition) patient stratification strategies, and feature reduction (denoising) [82,85]. More recently, Hi-C, a sequencing-based technique to detect the three-dimensional architecture of the nuclear genome, has been shown to effectively detect structural variants in B-cell acute lymphoblastic leukemia, a form of cancer that is frequently characterized by translocations [90]. Although promising, this approach is still non-standard in clinical practice and is currently used for research-only purposes.

### 4.3. Machine Learning Development Frameworks

The use of NGS technologies in clinical practice introduced the need for variant calling methods reproducibility and reusability. This favored the development of several dedicated open-source ML development libraries, and the diffusion of object-oriented programming languages in bioinformatics, including Python and R.

Convolutional kernels are the most exploited for variant calling [62,78], often combined with other architectures in ensemble methods [86]. Less frequently, other paradigms are used, including support vector machines (SVM) [91] and non-supervised learning [92], although they are generally restricted to a limited range of applications.

The landscape of ML-based variant discovery methods is dominated by deep neural networks, mainly due to the large amount of publicly available NGS data that can be used for training, validation/testing, and benchmarking against several gold standards (i.e., manually validated datasets with well-known outcomes), allowing these methodologies to outperform other competing methods (e.g., support vector machines, naïve Bayes, or random forests) [62]. This favored the creation of environments to easily develop custom NGS objects, methods, and tools for ML-based analyses.

TensorFlow [93] is the most used environment for the development of variant discovery AI-based software, followed by PyTorch [94].

Often, these environments make use of tools, such as Keras (url: https://github.com/fchollet/keras/ accessed on 12 July 2022) and Nucleus (https://blog.tensorflow.org/using-nucleus-and-tensorflow-for-dna.html accessed on 12 July 2022), offering a user-friendly experience and dedicated objects for sequencing data analysis. The bioinformatics community also uses the R environment, with dedicated packages and development tools, including the R port Torch [95]. However, R libraries are generally used for statistical computing, and currently could be less performant with respect to Python for ML development, which lists a much larger number of dedicated ML solutions.

## 5. Network-Based Approaches Applied to Cancer Research: Graph Theory and Causality for Analyzing the Biological Complexity

### 5.1. Graph Theory

Networks can be explored by the graph theory, useful to shed a light on their structure–function relationships [83]. In fact, graph-based approaches have been applied in extensive ways in different frameworks such as biology, chemistry, medicine, etc., [96] by providing a number of characterizations. Specifically, as described by Lecca et al., systems biology conceptualizes the networks of interacting molecules, and graph theory gives the mathematical tools to analyze them [96].

In particular, network analysis can use quantitative approaches to also model interactions between genes, proteins and other biological elements [97,98,99]. A general expression to refer to the investigation and modeling of these interactions is “molecular network”, which is becoming very important in cancer research as demonstrated in the applications against different types of neoplasm [100,101,102]: pancreas, gastric, lung, ovarian cancers and others are applications of graph theory in this framework.

Moreover, the graph theory allows us to decompose molecular networks in different subnetworks by directed subgraphs and multigraphs as demonstrated by Huang et al., modeling cancer networks, signal transduction networks, and cellular processes [100,101]. Over the years, different software has emerged in order to analyze the biological networks by the graph theory. A number of these are included in the R environment (https://www.R-project.org/ accessed on 12 July 2022), such as igraph (https://igraph.org/ accessed on 12 July 2022) [102], graph (https://github.com/ accessed on 12 July 2022), QuACN [103], network [104], Statnet [104] (https://statnet.org/ accessed on 12 July 2022) and NetBioV [105].

They are free packages that provide many functions to manage network systems, also by the Bioconductor platform (https://www.bioconductor.org/ accessed on 12 July 2022). These have many graphical functions, often inherited by the R environment, and the interesting advantage to follow object-oriented programming, that is suitable to use the elementary elements of a network in an independent and customized way. On the commercial side, the Dragon [106] software includes thousands of molecular descriptors to be used to analyze the biological network.

### 5.2. Causality

In the last decade, several methods have tried to model and quantify the causality in the biological and molecular networks, especially by considering the relationships among genes in a framework of perturbation of experiments and in presence of unfavorable factors. As a matter of fact, many phenomena in biology, medicine and other disciplines consider relationships among variables in a multivariate causal context. Hence, investigation and analysis of cause–effect relationships through statistical methods are incrementing, in order to explain how to test causal hypotheses, especially with a lack of randomized experiments [107]. Specifically, the methods try to translate the causal network into mathematical equations by generating assumptions on the nature -random or deterministic- of the variables (nodes of the network), and on the type -unidirectional or bidirectional- of the relationships (edges).

However, as suggested by Palluzzi et al. [108], a number of algorithms have been recommended to model and quantify causality in (biological/molecular) networks but they have low reproducibility and robustness, dependence on user-defined setup, and poor interpretability. In this framework, the structural equation models (SEM) provide a favorable methodology able to model and quantify the causality by an inferential approach, with an immediate and easy interpretation of the results [109]. In the SEM the relationships are assumed to be linear and the (response) variables supposed random are assumed to be multivariate normal. In the past few years, the SEM is catching on in cancer research as demonstrated by articles related to the modeling of the molecular networks in breast cancer [110], colorectal cancer [109], neuroblastoma [111] and leukemia, and more in general, in precision medicine [112]. At the same time, different packages emerged to analyze the causality of biological and molecular networks by the SEM. The majority is developed in the R environment. Firsts among everything, the lavaan [112] and SEMgraph [113] packages allow us to convert the causal diagram of the network into linear equations containing free (to be estimated) and fixed parameters. Of note, SEMgraph is a lavaan-based package that specifically manages complex biological systems as multivariate networks ensuring robustness and reproducibility through data-driven evaluation of model architecture and perturbation; that is readily interpretable in terms of causal effects among system components [108]. Finally, the other two R packages apply SEM specifically in a biological/molecular framework, GenomicSEM and GW-SEM [114], useful for modeling (i) the multivariate genetic structure of correlated traits by using a multivariate GWAS framework, and (ii) the associations of SNPs with phenotypes or hidden constructs on a genome-wide scale.

## 6. Homologous Recombination Deficiency: A New Bioinformatics Challenge

The complexity of biomarkers to be analyzed in clinical research and clinical practice is progressively increasing. This evolution also creates a new challenge for the bioinformatics analysis of sequencing data. As an example, in the following paragraphs, we describe the different strategies for the analysis of HRD, an emerging biomarker in oncology.

### 6.1. Testing Strategies for HRD Detection Based on Causes and Effects in the Genome

The homologous recombination repair (HRR) pathway is an error-free mechanism able to repair the DNA double-strand breaks (DSBs) during the S/G2 phases of the cell cycle [108,115]. In cells with a deficit of the HRR mechanism, the repair of DSBs occurs through alternative, error-prone methods, resulting in a high degree of genomic instability and in the accumulation of different alterations, including Single Nucleotide Variations (SNVs), small insertions and deletions (InDels), Copy number variation (CNV) or large-scale chromosomal rearrangements. Tumors with HRD are sensitive to poly (ADP-ribose) polymerase (PARP) inhibitors (PARPi), which suppress a second key DNA repair pathway of DNA single-strand breaks (SSBs) and create synthetic lethality in cancer cells with defects in HRR [116]. In particular, the PARPi Olaparib has been approved by the US Food and Drug Administration (FDA), in combination with bevacizumab, and by the European Medicines Agency (EMA), as a single agent, for the treatment of patients with advanced ovarian cancer associated with HRD-positive status and who are in complete or partial response to first-line platinum-based chemotherapy. FDA recently approved the PARPi Niraparib for the treatment of HRD-positive ovarian cancers, who have been treated with three or more prior chemotherapy regimens and who have progressed more than six months after responding to the last platinum-based chemotherapy [117,118,119]. In agreement with the FDA and EMA indications, the HRD status is defined by either a deleterious or suspected deleterious mutation in BRCA1 or BRCA2 genes, and/or genomic instability (GIS). To date, there are two principal strategies to identify tumors with HRD. The first strategy is focused on the identification of HRD causes using targeted sequencing with multi-gene panels able to evaluate alterations in the different genes of the HRR [120,121,122]. However, this approach has several limitations. Multi-gene panels are able to identify only the fraction of HRD cases related to genetic alteration of HRR genes. Indeed, the HRD might be caused by both genetic and epigenetic events. In addition, only some genomic alterations of HRR genes are associated with HRD [123]. Furthermore, the pathogenic role of many mutations of HRR genes is not known, due also to their very low frequency [124]. The second strategy is based on the study of the effects that HRD causes in the genome and looks for the genomic damage induced, independently from the originating mechanism [125,126,127]. These approaches vary from the analysis of genomic scars to the assessment of mutational signatures [128,129]. Genomic scars are the complex genomic alterations caused by HRD and represent a biomarker to identify patients who may benefit from treatment with PARPi [130]. Two commercial genomic scar assays have been developed to identify tumors with HRD, the “Myriad myChoice HRD” (Myriad Genetics; Salt Lake City, UT, USA) and the “FoundationOne CDx” (Foundation Medicine; Cambridge, MA, USA) tests.

The “myChoice HRD” assay is an NGS-based test able to detect variants in BRCA1 and BRCA2 genes and to determine a GIS score by measurement of three biomarkers: telomeric allelic imbalance (TAI), loss of heterozygosity (LOH) and large-scale transitions (LST) [131]. The HRD score is calculated by combining the LOH, TAI, and LST scores: tumors with a score ≥42 are classified as HRD-positive. This assay is the only one FDA approved for use in clinical practice [130]. The FoundationOne CDx is able to identify patients with HRD-positive status combining tumor BRCA1/2 mutational status with the rate of LOH [132,133]. The HRD score is measured as the percent of LOH in the tumor genome: genomic LOH ≥16% is classified as HRD-positive [132].

Despite the excellent results within several clinical trials, the HRD test based on genomic scar still has some technical limitations [134]. In addition, the presence of GIS based on a genomic scar can only indicate that at the time of testing the tumor had HRD. Indeed, the HRD scoring method is unable to account for reversion mutations that are predictive of platinum and PARPi resistance [135]. Conversely, newer approaches to HRD detection, including the identification of mutational signatures in sequencing data, potentially provide a dynamic readout of the current HRR status [136,137,138]. Multiple mutational processes generate a characteristic pattern of somatic mutations, termed “signatures” [139]. Multiple studies showed that one of these signatures, namely Signature 3 (Sig3), is associated with a deficiency in the HRR mechanism [140,141]. This mutational signature is based on SNVs and might only in part represent the complex genomic alterations associated with HRD. To further decode this complexity, a method based on the identification of signatures from copy-number (CN) features was developed. In the study by Macintyre et al., the CN signature 3, characterized by a distribution of breaks across all chromosomes and LOH, was significantly enriched in cases that displayed HRD caused by mutations in BRCA1/2 [142]. Moreover, signature 7 was associated with HRD and mutations in other HR genes, including BARD1, PALB2 and ATR, and loss of function mutations in PTEN [141]. Although these approaches, based on mutation or CNA signatures, have shown a good correlation with HR status, it is likely that a combination of different parameters can more accurately identify all cases with HRD [142].

### 6.2. Computational Tools for HRD Assessment

In recent years, machine learning algorithms and new computational tools have also been developed to perform a more complete and detailed analysis of the genomic alterations related to HRD, in order to better identify HRD-positive tumors [143,144,145]. Each algorithm has its own characteristics and specifications, applied to sequencing data from WGS, WES and TS. For the complete and detailed study of mutational signatures, it is possible to adopt different algorithms, such as HRDetect [146], Mutalisk [147], SigMA [148,149]. The application of each of these tools depends on the origin of the sequencing data, and therefore the available information.

HRDetect is primarily a mutational signature-based classifier designed to predict BRCA1 and BRCA2 deficiency based on six mutational signatures. It uses a lasso logistic regression model starting from sequencing data of WGS for HRD detection [146]. In particular, it allows us to calculate the HRD score recognizing the patterns of substitution base signatures and structural rearrangements. The HRDetect pipeline works on mutational data, such as: segments.tsv, somatic_indels.vcf, somatic_snvs.vcf, somatic_sv.tsv. The segments.tsv is used to identify and analyze the CNVs and the LOH score; while somatic_indels.vcf and somatic_snvs.vcf are used to study indels and SNVs in detail. Finally, the somatic_sv.tsv file is used to analyze the structural data of the variants. HRDetect has already been used on cohorts of patients with ovarian cancer, breast cancer and pancreatic cancer. The parameters used for HRD assessment include the evaluation of the main mutational signatures (such as Sig3), large deletions (>3 bp) with microhomology at the junction of the deletion, Rearrangement Signatures 3 and 5, and copy number profiles associated with widespread LOH [146]. The final output is a probability of BRCA1/2 mutation. The sensitivity and reliability of the results obtained with HRDetect changes according to the source of the data. By analyzing WGS data, the HRDetect reaches a sensitivity of 86%, setting the cut-off at 0.7 and the level of agreement at r = 0.96 as optimal parameters. By contrast, when HRDetect is applied to data obtained by WES, the sensitivity of detection is 46.8% [146,150].

To improve the results of HRDetect it is possible to use two different tools named Mutalisk and SigMA. Mutalisk (Mutation AnaLyIs ToolKit: www.mutalisk.org/ accessed on 12 July 2022) is an online computational framework used to investigate the signatures at a somatic level. This tool can be applied to genomic data generated by WGS, WES and TS sequencing using as input data the standard vcf file. There are two versions of Mutalisk, one is the web server and the other one is the R vs. 4.1.1. This algorithm identifies a maximum of seven mutational signatures at most from a specific somatic tissue [147]. For each signature set, a decomposition model can be generated using the maximum likelihood estimation method, or the multinomial test. It can be applied to HRD analysis for the identification of mutational signatures and for the classification of molecular processes mainly involved in the generation of pathogenic or benign mutations [147].

By contrast, SigMA (Signature Multivariate Analysis) algorithm performs a mapping of the most important mutational signatures from the SNV calls of WGS, WES or TS data associated with the HRD pathway [151]. This algorithm has a high sensitivity of 74% in identifying Sig3-derived rearrangements in HRD-positive tumors. The novelty of this algorithm is the application of the likelihood method, which allows us to associate a mutational spectrum to each patient [152].

Recently, a new tool named Classifier of Homologous Recombination Deficiency (**CHORD**) was developed (https://github.com/UMCUGenetics/CHORD/ accessed on 12 July 2022) for the detection of HRD status by BRCA1 and BRCA2 deficiency. CHORD is a random forest model used as a benchmark developed to detect pan-cancer HRD based on genome-wide mutational profiles using specific SNV, indels, and structural variants (SV) [153]. The CHORD algorithm uses deletions with flanking microhomology and 1–100 kb structural duplications to distinguish BRCA1-type HRD from BRCA2-type HRD [154]. The analysis is divided into two steps: the first step is based on the extraction of mutation contexts required by CHORD to create a matrix with all data. The second step is based on the prediction of HRD probabilities based on the calculation of an HRD score. Initially, this approach was used to calculate the HRD score in ovarian and breast cancer samples. Later, it was also extended to the analysis of other tumors in which BRCA1 and BRCA2 alterations are involved, such as pancreatic and prostate cancer [155] (Table 2).

Other more sophisticated and robust methods based on artificial intelligence have been introduced to identify and investigate the mutational signatures associated with HRD starting from WGS, WES and TS sequencing data. **PathAI** (https://www.pathai.com/ accessed on 12 July 2022) employs machine learning models that predict the HRD status by studying how the disease evolves and making dynamic models of it using the mutational signatures. The **GSA** (genomic scar analysis) algorithm was developed and vali. dated to calculate the HRD score and the LOH score [156]. This approach is characterized by the presence of two submodules: tree recursion (TR) segmentation and filtering, and the estimation and correction of the tumor purity and ploidy. These elements are important for a better analysis of the HRD/LOH score. The input data formats of GSA are (i) BAF data and (ii) LRR data. BAF (B allele frequency) represents the median SNP genotype frequency of each capture region while LRR (Log R ratio) is the normalized depth ratio of the tumor and the normal sample (or blood cell control set) in each capture region after GC-bias correction. Currently, none of the above-described gene signatures have been widely adopted in clinical practice because they were identified based solely on a single dataset and did not take into consideration the heterogeneity of patient cohorts [158]. A recent approach under investigation, called **AcornHRD** (https://ascopubs.org/ accessed on 12 July 2022), enables the calculation of an HRD score associated with the efficacy of PARP inhibition and platinum-based chemotherapy in a variety of cancer types. The aim of this approach is to extend the range of patients that might benefit from targeted therapy. A current limitation of the above-described methods is the impossibility to capture tumor evolution processes, such as a restoration of HRR function in response to therapy-selective pressure. Therefore, it could be useful to incorporate functional biomarkers based on dynamic changes in DNA repair that occur throughout tumor evolution for the identification of HRD-positive tumors [157,159].

## 7. Discussion

The use of NGS technologies in clinical practice and clinical research is progressively increasing. The guidelines of the main scientific societies recommend the use of NGS in the diagnosis of numerous human cancers [160,161]. Furthermore, the availability of clinical studies for often rare and complex genomic alterations requires the use of large TS panels to facilitate the enrollment of patients in studies with new drugs. In this complex scenario, in which the therapeutic decision depends mainly on the genomic landscape of the tumor of each individual patient, the quality and accuracy of the NGS analysis are essential to guarantee the appropriateness of the treatments. Therefore, having a robust, reliable and validated bioinformatics pipeline available is a necessary requirement to be able to analyze genomic data and provide useful results for the clinical decision.

The introduction first in research and then in clinical practice of complex genomic markers, such as MSI, TMB and HRD, has made the analysis of the sequencing data even more complex. As we have discussed for HRD, it is essential to identify bioinformatics tools capable of deriving these complex biomarkers also from TS data, in order to favor the implementation of these new biomarkers in the field of diagnostics and clinical research, with sustainable costs and times and methods of analysis compatible with clinical needs.

All studies so far have evaluated the correlation between genomic instability and response to platinum and/or PARPi. However, genomic instability could also represent an important marker of response to immunotherapy, as suggested by preliminary data [162]. Studies in this direction are certainly needed. It is clear that HRD plays a crucial role in cancer pathogenesis and progression. Hence, accurate estimation of HRD status is essential, not only to guide treatment decisions but also for the development of novel therapeutic strategies, with the ultimate objective of expanding the pool of patients who may derive clinical benefit from such approaches. Therefore, there is an urgent need to further develop reliable HRD detection methodologies that are comprehensive, cost-effective, and minimally invasive with a high predictive value for treatment response and disease progression [163].

## 8. Conclusions

In conclusion, the field of genomic biomarkers in oncology is constantly evolving and we expect that they will become increasingly important for precision oncology. The use of AI techniques not only for the interpretation of these data but also for their integration with the clinical and pathological characteristics of the patient represents a future challenge for cancer research.

## Figures and Tables

**Figure 1 biomedicines-10-02074-f001:**
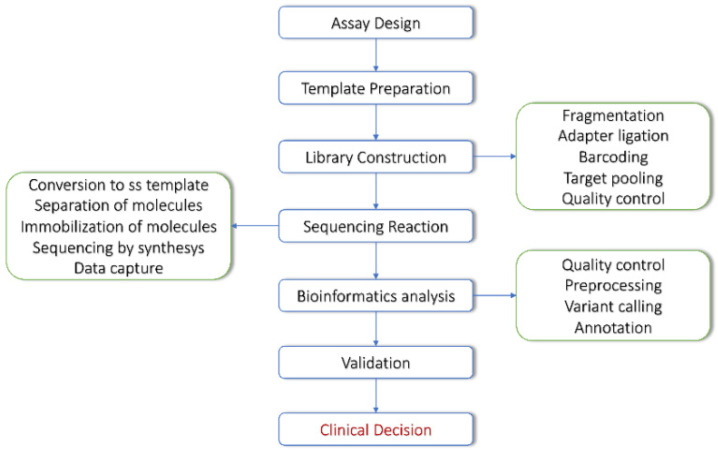
NGS workflow.

**Figure 2 biomedicines-10-02074-f002:**
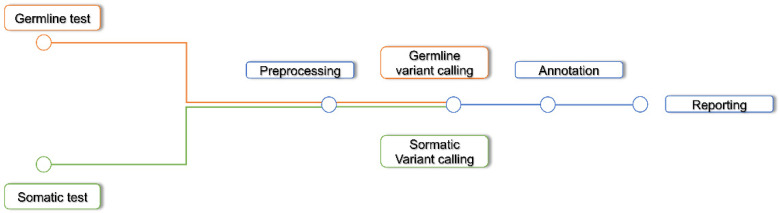
Nf-core Sarek pipeline.

**Table 1 biomedicines-10-02074-t001:** Tools included in the nf-core Sarek framework. The table reports the tool name, the sample type (G: germline, S: somatic), the variant type (SNV: small nucleotide variant, indel: small insertion/deletion, SV: structural variant, CNV: copy number variant, MSI: microsatellite instability), a small description of the core method, and the latest literature reference link.

Tool	Sample Type	Variant Type	Method	Ref
Manta	G, S	SV, indels	Graph-based breakend analysis	[57]
TIDDIT	G, S	SV	Coverage-based genome scan	[58]
Cnvkit	G, S	CNV	Coverage-based genome scan	[59]
Freebayes	G, S	SNV, indels	Haplotype-based Bayes theorem	[60]
Strelka2	G, S	SNV, indels	Haplotype-based mixture modeling	[61]
DeepVariant	G	SNV, indels	Pileup image CNN classification	[62]
HaplotypeCaller	G	SNV, indels	Haplotype re-assembly, likelihood	[63]
Mpileup	G	SNV, indels	Local re-alignment, likelihood	[55]
Mutect2	S	SNV, indels	GATK + read-to-haplotype alignment	[64]
Ascat	S	CNV	Signal intensity and allele frequency	[65]
Control-FREEC	S	CNV	LASSO-based genome segmentation	[66]
MSIsensor-pro	S	MSI	Multinomial distribution	[67]

**Note.** G: germline, S: Somatic; SV: Structural variant; SNV: small nucleotide variant, indel: small insertion/deletion; CNV: copy number variant, MSI: microsatellite instability.

**Table 2 biomedicines-10-02074-t002:** HRD computational tools.

Tools	Applications	Variants Type	References
HRDetect	WGS	indels, snv, sv and CNV	[146]
Mutalisk	WGS, WES and TS	Mutational signatures	[147]
SigMA	WGS, WES and TS	SNV	[148]
CHORD	WGS	SNV, indels	[153,154]
PathAI	WGS, WES and TS	Indels, snv	https://www.pathai.com (accessed on 12 July 2022)
GSA	WGS, WES and TS	CNV	[156]
AcornHRD	WGS, WES and TS	Indels, CNV and snv	[157]

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
