# Peer review of "Bioinformatics: From NGS Data to Biological Complexity in Variant Detection and Oncological Clinical Practice"

_biomedicines, 2022, doi:10.3390/biomedicines10092074_

Round 1

Reviewer 1 Report

Attached below

Author Response

Reviewer comment 1.

First is the description of bioinformatic tools and pipeline for variant calling process. This section is written in a very crude manner and not updated to 2022. The description feels like reading about variant calling in early 2010s. The authors need to seriously update this section to properly explain the variant calling steps and describe the algorithms behind commonly used variant calling methods like Mutect2, etc.

Answer:

We thank the Reviewer for this comment. We globally improved sections 1 and 2, to provide a stronger and more gradual introduction to NGS technologies and their role in clinical practice. Following Reviewer’s suggestions, we focused our attention on variant calling and related pre-processing steps.

English has been improved as well: we corrected grammatical and syntactic errors, and rephrased misleading sentences, according to the Reviewer’s suggestions.

We also added a description of the state-of-the art tools for variant calling in clinical practice (section 3.5 Variant calling), with a new table reporting name, input sample, output variant type, core method, and latest publication for each of them (Table 1).

We would also like to point out the critical role of the nf-core/Sarek project in unifying and standardizing the pre-processing and variant calling methods in the last years, and how this was made possible through the container technology. This is a recent advancement in the field and we extensively explained it through sections 3.5 and 3.6.

Although relatively new, Sarek includes tools that are widely used by the bioinformatics community, based on local read (re-)alignment, coverage analysis and/or haplotype determination (phasing), followed by likelihood-based testing. The only machine learning (ML)-based tool in this standard list is Google’s DeepVariant. This because ML-based approaches are still non-standard in clinical practice, although extensively used in cancer research. Nevertheless, section 4 lists and describes the most recent advancements in the field of ML for variant calling and its future direction.

Reviewer comment 2.

There are more sequencing technologies like RNA-Seq and several other epigenetic sequencing technologies that have been around for close to a decade and are being introduced in the clinical practice especially in detecting the higher-level molecular interactions that shape the complexity of cancer. This paper does not comment on these technologies and hence the authors should limit the scope change their title and indicate in the Abstract that this only covers variant calling. Unless the authors address serious deficiencies in description of variant calling pipeline (I have listed some specific points below with line numbers), I do not recommend publication.

Alternatively, the authors can limit the scope of their review and make this review a summary of computational algorithms used in addressing the biological challenges of analyzing NGS data to unravel complexity of cancer. I would still recommend that authors replace first section with just the description of different variant calling algorithms without going in detail about different steps for the bioinformatics pipeline. That will make this paper a very effective summary of the computational algorithms.

Answer:

We thank the Reviewer for this comment. We restructured the introduction and section 2 to better introduce variant calling, that is the focus of our review. We modified title, abstract, and keywords to clearly focus the attention on variant calling-related NGS technologies and tools.

In addition, variant calling sections have been improved according to the Reviewer’s suggestions, as reported in the other answers.

Line 32-33. The statement is grammatically incorrect: “intention of elaborate processing data …”

The statement was corrected in line 43-44 in the track changes revised manuscript.

Line 34. The statement start is grammatically incorrect: “Despite what?”

The statement was corrected in line 45 in the track changes revised manuscript..

Line 45: Point 1 title. I would recommend better usage of words. This seems not professional for publication in a scientific journal.

The introductory chapters 1 and 2 have been extensively restructured, including titles.

Line 57: The statement starts well but the second half does not make sense. An important phrase is missing. What plays an important role in the massive parallel sequencing? NGS itself is massive parallel sequencing but it is not clear what plays an important role.

This sentence has been rephrased (section 2.1 NGS approaches in cancer patients’ management).

Line 62-63: The sentence is misleading. NGS is generic term for a set of different sequencing experiments that have different scope of coverage across the genome from a targeted set of genes to the whole genome.

We agree with the reviewer, and we completely rearranged that sentence and the following part of the paragraph.

Line 65-67: I would recommend authors to indicate that protein-coding genes only constitute 3% of the human genome. Hence WES reduces costs substantially compared to WGS.

We agree with the reviewer. We added it to the main text (section 2.1 NGS approaches in cancer patients’ management).

Line 70-72: Probably a good term will be to say that “it validated lot of ...alterations that were previously identified using other methods”.

Line 70-72: The cited references do not indicate much on the validated alterations. The authors should cite more specific cases where NGS variant calling validated previously known alterations.

We rephrased the sentence and we added three new references (section 2.1 NGS approaches in cancer patients’ management).

Line 158-159: As authors said, there are many tools out there for variant calling. The authors should describe some of the them or list their categories based on what kind of algorithms they work (Bayesian, probabilistic, etc).

Figure 2 is highly misleading. You cannot know whether an identified variant is somatic or germline even before the pipeline has started. This step comes at the annotation level. If the authors indicate that there are two different pipelines: one for germline and another for somatic, this figure does not show any difference between the two. Please correct this

figure to show the steps more illustratively.

Answer:

We added a description of the state-of-the art tools for variant calling in clinical practice (section 3.5 Variant calling), with a new table reporting name, input sample, output variant type, core method, and latest publication of the algorithm (New Table 1) for each of them.

We would also like to highlight the central role we gave to emerging machine learning-based approaches. Although still non-standard in clinical practice, machine learning could better take advantage of the large amounts of available NGS data to improve detection accuracy, overcoming common issues such as high-dimensionality and non-linearity. Furthermore, they may tackle new aspects of variant calling (e.g., chromatin state-based noncoding variants detection), generally disregarded by classical approaches.

This is extensively discussed throughout section 4.

We also improved figure 2 (section 3.6 Variant discovery pipelines: tools and algorithms), to clarify the different steps of nf-core/Sarek variant calling process and how the germline/somatic distinction can be clear from the beginning, depending on the sample type and required test (e.g., blood for germline and/or solid tumour tissue for somatic).

Line 133-134: “... genomics genomic variants.”

The error was corrected (section 3.4 Pre processing)

Line 290-292: The authors talk about use of ML algorithms to predict chromatin state and accessibility but do not comment on how this information is learned by the ML algorithm. The authors should highlight the Hi-C and methylation based NGS technology that gives

information on epigenetic interactions and state of chromatin.

We thank the Reviewer for this suggestion. We added further details (section 4.2 ML and deep neural network for variant discovery)

Line 299-300: Statement grammatically incorrect.

This sentence was rephrased (section 4.3 Machine learning development frameworks).

Line 624: Journal name is missing. Please correct the citation.

The reference was fixed.

Reviewer 2 Report

Datolo et al discuss in this review how the bioinformatics field helps interpret NGS data in cancer, including using machine learning algorithms. The work has a broad introduction explaining what next-generation sequencing is and which techniques are used, focusing on genomics data, WES and WGS. They included as well a very explicative NGS experiment workflow. The work also mentions the steps of variant discovery and the software associated with this step. After the variant calling path, the review shows machine learning methods that are used for cancer research, including those used with python and R. 

The review brings essential information in the cancer research area in variant discovery and the contextualization using machine learning with this type of data. However, there are some critical points to improve:

. The title "Bioinformatics: from NGS data to biological complexity in cancer" is misleading, when we talk about NGS , Next Generation Sequencing, we are talking about multiple types of sequencing methodologies, including transcriptomics (different RNA-Seq methodologies) and epigenomics approaches (such as Methylation methods, Chip-seq, ATAC-seq) (Levy and Boone, 2019 - doi: 10.1101/cshperspect.a025791 ; Goodwin et al., 2016 - doi: 10.1038/nrg.2016.49). Unless the authors decide to include these order approaches, the title must change to be more specific to include genome/exome sequencing. The sentence that starts into line 26 also misleads to a wrong concept of NGS . 

"The need to identify an increasing number of complex genomic biomarkers has led to the introduction in clinical practice of technologies based on massive parallel sequencing, more commonly known as next generation sequencing (NGS). "

Here it is also confusing if genomic massive parallel sequencing is the same that Next Generation Sequencing. And in the line 61 there is another sentence that brings confusion about the term: 

"Importantly, NGS is a generic term that refers to different approaches that are able to cover from a few genes to the whole genome [20–22]".   

Please state clearly the difference between the term "DNA sequencing" (both WGS and WES)  and NGS. 

. The review misses a clear separation of the topics, and there is not a pattern in the topic separation. It starts with "1.Introduction" and the second topic is: "point one Point 1. From wet lab to web lab technologies to analyze NGS raw data NGS approaches in cancer patients' management". I recommend reading other review publications available in the journal such as Zakrzewicz  and Geyer, 2022 to improve topic organization. It should be such as 1. Introduction, 2. ,2.1,2.2,3. ,etc.

. Figure 2 is a remodeled figure from Garcia et al,  2020 remodeled. The authors can create a figure that compares different software or even a table showing a difference, but a figure of a published pipeline cannot be a review publication figure. 

. The "conclusions" topic needs to be separated into discussion (Unmet needs and future perspectives) and conclusion which is the last paragraph.

Author Response

REVIEWER 2

Reviewer comment 1.

The title "Bioinformatics: from NGS data to biological complexity in cancer" is misleading, when we talk about NGS , Next Generation Sequencing, we are talking about multiple types of sequencing methodologies, including transcriptomics (different RNA-Seq methodologies) and epigenomics approaches (such as Methylation methods, Chip-seq, ATAC-seq) (Levy and Boone, 2019 - doi: 10.1101/cshperspect.a025791 ; Goodwin et al., 2016 - doi: 10.1038/nrg.2016.49). Unless the authors decide to include these order approaches, the title must change to be more specific to include genome/exome sequencing. The sentence that starts into line 26 also misleads to a wrong concept of NGS .

Answer:

We thank the reviewer for this comment. We improved the title to clearly focus on variant detection in oncological clinical practice.

We also corrected the misleading sentence and rearranged the introduction accordingly.

Reviewer comment 2.

"The need to identify an increasing number of complex genomic biomarkers has led to the introduction in clinical practice of technologies based on massive parallel sequencing, more commonly known as next generation sequencing (NGS). "

Here it is also confusing if genomic massive parallel sequencing is the same that Next Generation Sequencing. And in the line 61 there is another sentence that brings confusion about the term: 

"Importantly, NGS is a generic term that refers to different approaches that are able to cover from a few genes to the whole genome [20–22]".   

Please state clearly the difference between the term "DNA sequencing" (both WGS and WES) and NGS.

Answer:

We thank the reviewer for this comment.  We rephrased the entire section and rearranged section 1 and 2 to better clarify these concepts.

Reviewer comment 3.

The review misses a clear separation of the topics, and there is not a pattern in the topic separation. It starts with "1.Introduction" and the second topic is: "point one Point 1. From wet lab to web lab technologies to analyze NGS raw data NGS approaches in cancer patients' management". I recommend reading other review publications available in the journal such as Zakrzewicz  and Geyer, 2022 to improve topic organization. It should be such as 1. Introduction, 2. ,2.1,2.2,3. ,etc.

Answer:

We reorganized all sections according to the reviewer suggestions.

Reviewer comment 4.

Figure 2 is a remodeled figure from Garcia et al,  2020 remodeled. The authors can create a figure that compares different software or even a table showing a difference, but a figure of a published pipeline cannot be a review publication figure. 

Answer:

We agree with the reviewer. We changed figure 2 accordingly.

Reviewer comment 5.

The "conclusions" topic needs to be separated into discussion (Unmet needs and future perspectives) and conclusion which is the last paragraph.

Answer:

We agree with the reviewer. We divided the last paragraph in discussion and conclusions.

Round 2

Reviewer 1 Report

The authors have significantly improved their writing and the content for the first topic. I believe with the extensive corrections, the review manuscript is ready for publication

Reviewer 2 Report

The authors corrected all the demands that I requested.